# Spatio-Temporal Variation and Influencing Factors of the Coupling Coordination Degree of Production-Living-Ecological Space in China

**DOI:** 10.3390/ijerph191610370

**Published:** 2022-08-20

**Authors:** Xinghua Cui, Ning Xu, Wanxu Chen, Guanzheng Wang, Jiale Liang, Sipei Pan, Binqiao Duan

**Affiliations:** 1School of Economics, Jiangxi University of Finance and Economics, Nanchang 330013, China; 2School of Political Science and Public Administration, Henan Normal University, Xinxiang 453007, China; 3School of Geography and Information Engineering, China University of Geosciences (WuHan), Wuhan 430074, China; 4School of Geography and Oceanography Sciences, Nanjing University, Nanjing 210023, China; 5College of Public Administration, Nanjing Agricultural University, Nanjing 210095, China

**Keywords:** production-living-ecological space, gravity center analysis, spatial autocorrelation, coupling coordination degree model, geographically weighted regression model, influencing factor, China

## Abstract

Territorial space is a multi-functional complex. The coordinated production-living-ecological space (PLES) effectively coordinates the man-land relationship, promotes regional sustainable development, and maximizes territorial space. How to build a high-quality national spatial layout and support system for development has become a hot topic of concern in all sectors of society. However, few studies have explored the coupling coordination considering the various production-living-ecological functions of land use type and its influencing factors of PLES at the county scale in China. To address the gap, based on the connotation of PLES theory, this study established a classification and evaluation system for PLES and analyzed the spatio-temporal characteristics, coupling coordination degree, spatial autocorrelation, and influencing factors of PLES in China from 2000 to 2020. The results are as follows: (1) The production space index and living space index in China showed a continuous increase tendency, while the ecological space indexs decreased continuously during the study period. The production space and living space were concentrated in the east of Hu Line, and the ecological space indexes in mountainous areas were significantly higher than those in plain areas during the study period. (2) The gravity centers of PLES all migrated to the west of China to different degrees during the study period. (3) From 2000 to 2020, the basically balanced category was the main coupling coordination type, and the number of seriously unbalanced categories accounted for the least. In the west of the Hu Line, the seriously unbalanced category was dominant, while in the east of the Hu Line were the moderately unbalanced categories and above. (4) During the study period, the low-low type was the main relationship type, widely distributed in western China, followed by the high-high type, mainly situated in the North China Plain, Yangtze River Delta, Pearl River Delta, Jianghan Plain, Chengdu Plain, Northeast China Plain, and some provincial capital cities. (5) Regression results showed that natural factors were the main reason restricting the coordinated development of PLES, and socioeconomic factors could effectively promote the coordinated development of PLES. Landscape pattern also significantly influenced the coordinated development of PLES, but varied greatly. The findings of this study can provide a scientific reference for the optimization of territorial space layout and the promotion of high-quality development of territorial space.

## 1. Introduction

Since the 21st century, China’s socioeconomic development has undergone a profound transformation, and its territorial spatial pattern, form, function, and ecological environment are undergoing profound reconstruction [1,2,3]. The rapid advancement of urbanization has intensified the demand for production-living-ecological space (PLES). The conflict of PLES has led to a series of issues, including out-of-control space development, disordered regional competition, tightening resource and environmental constraints, a deteriorating human-capital relationship, and serious damage to the ecological environment [4,5,6]. Constructing a new pattern of territorial space development and protection, gradually forming a PLES with coordinated functions and a new pattern of territorial space development and protection with obvious main functions, complementary advantages, and high-quality development are effective ways to alleviate the multiple problems facing China’s territorial space utilization [7,8,9]. Coordinating the functional coupling and coordinated development level of PLES is the inherent requirement of high-quality utilization for territorial space [2,10,11]. However, existing research is mostly based on the land use type area to measure PLES, and there is a lack of research on assessing PLES through production-living-ecological functions of land use type. Therefore, scientific measurement of the spatio-temporal evolution characteristics and the influencing factors of the coupling coordination relationships of PLES in China is an important reference for constructing the spatial layout and supporting system of high-quality development.

The emergence of PLES has attracted the attention of researchers. Significant research has been carried out in various fields, and remarkable progress has been made in the study of PLES. However, there are still deficiencies in the existing research. First, partial studies only measure PLES without further exploring its coupling coordination. Second, although some studies have explored the coupling coordination of PLES, they are mostly based on the index system or the land use type area to estimate PLES [2,11,12], and rarely consider the production-living-ecological functions of the land use type. Third, at present, there is no research on the influencing factors of coupling coordination of PLES based on production-living-ecological functions of the land use type. Lastly, research on PLES at the county scale in China is still insufficient. Considering the above, this study intends to measure PLES and introduces the physical coupling model to further explore the coupling coordination of PLES based on production-living-ecological functions of land use type to fill the gap. Meanwhile, three factors including the natural environment, social economy, and landscape pattern are selected to explain the influencing factors of coupling coordination of PLES.

With the acceleration of China’s industrialization and urbanization, the spatial structure of PLES has become unbalanced, resulting in intensified conflicts of the spatial structure of PLES [13], which reflects the disorder of land use in China and restricts regional sustainable development, causing a series of social problems [2]. The Third Plenum of 18th Central Committee of Communist Party of China proposed that the future development direction of land resources should gradually change to the mode of coordinated development of PLES [14]. Exploring influencing factors of coupling coordination of PLES is not only in line with the current national policy on PLES, but also can provide scientific references for the coupling coordination of PLES. Therefore, based on the classification and evaluation system of PLES in China, this study measured the spatio-temporal differentiation characteristics of the PLES index at the county scale in China based on land use remote sensing monitoring data in 2000, 2010, and 2020, and analyzed the gravity centers migration trajectory of PLES using gravity centers analysis method. Then, we further measured the coupling coordination degree of PLES and analyzed the spatial autocorrelation of the coupling coordination degree of PLES. Finally, the spatial non-stationary influencing factors of the coupling coordination degree of PLES in China were analyzed by the geographically weighted regression (GWR) model. It is expected that this study can provide scientific support for promoting the coordinated development of PLES, constructing the development and protection pattern of territorial space in the new era, and guide the high-quality development of territorial space. Specifically, the study aims to: (1) Reveal the spatio-temporal distribution characteristics of China’s PLES index. (2) Evaluate the characteristics of the PLES coupling coordination index in China. (3) Analyze the influencing factors of the PLES coupling coordination degree in China.

## 2. Literature Review

### 2.1. PLES

The coupling coordination of PLES is an extension of the connotation of coordinating the man-land relationship and promoting sustainable development [2,10,11]. Previous studies on PLES mainly focused on PLES classification [15], index construction [2,16,17,18], pattern evolution [19,20,21], spatial conflict [4,5], spatial optimization [8,9,16,22,23], eco-environmental effect [6,24,25], and coupling coordination analysis [2,11,12]. Specifically, regarding the classification of PLES, previous scholars have completed abundant studies on land classification and evaluation systems based on different research perspectives [2,17,26]. For example, by emphasizing the concept of ecological land use and starting from the main function of land, the classification system of PLES is constructed, which mainly includes ecological land use type, eco-production land use type, production ecological land use type, and living and production land use type [27]. By quantifying the value of the PLES function, some studies have constructed a functional classification system of PLES from the perspective of ecosystem services, landscape functions, and land functions, and achieved quantitative identification of the dominant functions of PLES [28]. In addition, a large number of studies have established the classification and evaluation system of PLES in China based on the dialectical relationship between land use function and land use type by emphasizing the connection between PLES and existing land use classification standards [2,20,29].

Using the classification system of PLES based on multi-function division of land use, previous studies have undertaken a mass of work on PLES change, spatial conflict, spatial optimization, and eco-environmental effect [2,4,20,30,31]. For example, Yuan et al. adopted the land use transfer matrix method to analyze the spatio-temporal variation characteristics of the quantity and structure of PLES to identify different types of spatial conflict problems [32,33]. Li et al. combined the land use transfer matrix with a standard deviation ellipse and other mathematical analysis methods to discuss the evolution of the PLES pattern and its influencing factors [2,17]. The influence intensity of different factors on the change of the PLES spatial pattern is quantified, and it is pointed out that natural factors significantly limit territorial spatial patterns [1,33]. In terms of the eco-environmental effects of PLES, some studies combined the evolution characteristics of PLES with eco-environmental quality, quantified the eco-environmental effects brought by the evolution of PLES pattern, and clarified the impact of PLES transformation on eco-environmental quality [6,24,31]. In addition, previous literature based on the comprehensive perspective of PLES evaluated the function index of PLES and analyzed the coupling coordination of PLES by constructing the index system of PLES, providing ideas for optimizing the spatial layout of PLES [2,11,12].

### 2.2. Driving Forces of Coupling Coordination Degree of PLES

Currently, extensive studies have explored the influencing factors of PLES from different perspectives: human factors (e.g., GDP, population density), natural factors (e.g., annual average temperature and annual average precipitation), and land use types (e.g., cultivated land area, urban land area). Deng et al. measured the influencing factors of the spatial evolution of PLES [17]. Moreover, research also explores ecological environment effects and its influencing factors of land use change based on the viewpoint of PLES from the natural environment (e.g., slope, annual average precipitation) and social economy (e.g., population density, land use intensity) [31]. In addition, similar to our study, altitude, slope, temperature, precipitation, population, and economic factors were selected as independent variables to explore their influences on the coupling coordination of PLES by Li et al. [2]. In previous studies, research methods mainly include multiple linear regression, spatial autocorrelation, and the GWR model [2,17,31]. Specifically, the GWR model is widely used to explore the influencing factors of the coupling coordination of PLES because it almost eliminates spatial autocorrelation [34,35]. However, current studies on influencing factors of coupling coordination of PLES are all based on the land use type area, which cannot fully reflect the production-living-ecological functions of various land use types [34,35]. Therefore, it is necessary to measure the coupling coordination of PLES based on the production-living-ecological land use classification and assessment, which can provide scientific references and effective guidance for the pattern optimization of PLES.

## 3. Materials and Methods

### 3.1. Data Sources

Land use/land cover change datasets in China in 2000, 2010, and 2020 are obtained from Resources and Environmental Science and Data Center, Chinese Academy of Sciences (RESDC) (http://www.resdc.cn (accessed on 17 August 2022)), with a resolution of 1000 m × 1000 m [36]. It has the highest accuracy among the land use remote sensing monitoring data products in China. China’s multi-period land use/land cover remote sensing monitoring dataset is based on the Landsat remote sensing image data of the United States as the primary information source and obtained by manual visual interpretation. The data for 2000 and 2010 are mainly based on Landsat TM/ETM remote sensing images of the same period as the main data source, and the land use data for 2020 is conducted based on the land use data of 2015 through manual visual interpretation of Landsat 8 remote sensing images. The land use types cover 7 first-level land use types (e.g., cultivated land, forestland, grassland, water area, construction land, wetland, and unused land) and 25 second-level land use types. In 2010, the accuracy of first-level land use data was higher than 94.3%, and the overall accuracy of second-level land use data was higher than 91.2% [37]. Additionally, climate and topographic data with a resolution of 1000 m also come from the RESDC. Resolution 100 m population data is derived from the WorldPop (https://www.worldpop.org/ (accessed on 17 August 2022)), and Arc Toolbox/Spatial Analyst Tools/Zonal/Zonal Statistics of ArcGIS10.3 were used to extract data and finally achieve the quantification and spatialization of relevant influence factors. Taiwan, Hong Kong, and Macau in China were not included in this study due to data limitations.

### 3.2. Methods

#### 3.2.1. Definition and Calculation of PLES Index

Land is a multi-functional complex, which can perform multiple land use functions. However, due to different land use methods, intensification, and related users, land can show the priority and level of production, living, and ecological functions [29,38]. The construction of the land use classification system of PLES is the premise of analyzing the PLES pattern. Based on the research of Liu et al. and Cui et al. [29,39], this study quantified the various functions provided by different land use types, as shown in Table 1. Besides, production space index (PSI), living space index (LSI), and ecological space index (ESI) were constructed to measure the PLES index level. The calculation equations are as follows:(1)$$PSI=\sum_{i=1}^{n}\left(LA_{i,t}\times PS_{i,t}\right)/\sum_{i=1}^{n}LA_{i,t}$$
(2)$$LSI=\sum_{i=1}^{n}\left(LA_{i,t}\times LS_{i,t}\right)/\sum_{i=1}^{n}LA_{i,t}$$
(3)$$ESI=\sum_{i=1}^{n}\left(LA_{i,t}\times ES_{i,t}\right)/\sum_{i=1}^{n}LA_{i,t}$$
where *LA_i,t_* represents the area of land use type *i* at time *t*; n represents the number of land use types in this study; *PS_i,t_*, *LS_i,t_*, and *ES_i,t_* represent the production land, living land, and ecological land score of land use type *i* at time *t* in Table 1, respectively.

#### 3.2.2. Gravity Centers of PLES Index

The center of gravity transfer model has been applied to geographical research in recent years, which can intuitively reflect the continuous change trajectory of various types of land space at a certain time. To date, significant studies have introduced the gravity center transfer model to analyze the center of gravity transfer pattern of PLES [40,41]. Therefore, to analyze the gravity center migration trajectory of PLES index in China at the county scale, this study measured the gravity center evolution law of China’s PLES index in 2000, 2010, and 2020 with the gravity center analysis model [42]. The calculation equations are as follows:(4)$$X_{t}=\sum_{j=1}^{n}PLES_{tj}X_{j}/\sum_{j=1}^{n}PLES_{tj}$$
(5)$$Y_{t}=\sum_{j=1}^{n}PLES_{tj}Y_{j}/\sum_{j=1}^{n}PLES_{tj}$$
where *X_t_* and *Y_t_* denote the longitude and latitude coordinates of gravity center of PLES index, respectively; *X_j_* and *Y_j_* denote the longitude and latitude coordinates of the gravity center of unit *j*, respectively; *PLES_tj_* denotes the index of PSI, LSI, or ESI of unit *j* at time *t*.

#### 3.2.3. Coupling Coordination Analysis of PLES Index

The coupling degree of the PLES index (C) characterizes the degree of mutual promotion, mutual influence, and mutual coercion among PLES. In this study, a physical coupling model was introduced to construct a dynamic coupling degree model and coupling coordination degree model among PSI, LSI, and ESI, respectively, to quantitatively measure the coupling process and evolution trend of PLES at the county scale in China [2,11,12,43]. The calculation equation is as follows:(6)$$C={\left\{PSI\cdot LSI\cdot ESI/\left({\left(PSI+LSI+ESI/3\right)}^{3}\right)\right\}}^{1/3}$$

The coupling degree can only reflect the correlation degree among PLES, but can’t determine the types of mutual promotion or restriction. It may occur that the values of the three types of PLES are low but the coupling degrees are high. To further analyze the coordinated development degree of PLES from disorder to order on the basis of mutual coupling, this study constructed the coupling coordination degree model (D) based on the coupling degree. The calculation equations are as follows:(7)T=αPSI+βLSI+λESI
(8)$$D=\sqrt{T\cdot C}$$
where the value of *C* is [0, 1], the larger the value is, the stronger the interaction among PLES is. The value of *D* is also [0, 1], and the larger the value, the higher the coupling coordination level. *T* represents the comprehensive harmonic index of PLES. *α*, *β*, and *γ* represent the contribution share of PLES. In this study, PLES contributes equally to the coordinated development of the whole (*α* = *β* = *γ* = 1/3). Based on the calculation results, the coupling coordination degree of PLES was divided into five categories: seriously unbalanced (0–0.20), moderately unbalanced (0.21–0.40), basically balanced (0.41–0.60), moderately balanced (0.61–0.80), and highly balanced (0.81–1) [43]. Seriously unbalanced indicates that the excessive development of production function has led to the serious extrusion of living and ecological space, resulting in a series of ecological problems. Moderately unbalanced presents that the production function still occupies an absolute dominant position and the living function is gradually improved. While the ecological problems caused by domestic pollution are gradually prominent. Basically balanced shows that the speed of regional development has slowed down, and it has gradually transformed into an intensive and efficient production mode. Moreover, it has begun to pay attention to the restoration of ecological problems caused by living and production activities. Moderately balanced indicates that certain achievements have been made in ecological restoration, and the overall living environment has been greatly improved. Highly balanced explains that the functions of PLES promote each other, which can satisfy the needs of different stakeholders and realize the effective development of spatial systems [44]. Specific classification methods are shown in Table 2. Moreover, to further reflect the coupling coordination of subsystems of PLES, we also used the formula model to measure the coupling coordination degree between production space and living space, living space and ecological space, and production space and ecological space, respectively.

#### 3.2.4. Spatial Autocorrelation

In this study, we introduced exploratory spatial data analysis to reveal the spatial aggregation features of coupling coordination of PLES [33,34]. The global Moran’s Index was used to measure the overall distribution characteristics of coupling coordination of PLES. The calculation equation is as follows:(9)$$I=\frac{n\sum_{i=1}^{n}\sum_{i=1}^{n}\omega_{ij}\left(x_{i}-\bar{x}\right)\left(x_{j}-\bar{x}\right)}{\sum_{i=1}^{n}\sum_{i=1}^{n}\omega_{ij}\sum_{i=1}^{n}{\left(x_{i}-\bar{x}\right)}^{2}}$$
where *x_i_* is the observation value for the *i*-cell attribute value. (*x_i_* − $\bar{x}$)(*x_j_* − $\bar{x}$) reflects the similarity of observed values. $\omega_{ij}$ is the spatial weight matrix between the spatial units *i* and *j*.

Local spatial autocorrelation can characterize the similarity degree of spatial units, indicating how spatial dependence changes with spatial location. The calculation equation is as follows:(10)$$I_{i}=\frac{x_{i}-\bar{X}}{S_{i}^{2}}\sum_{j=1,i\neq j}^{n}\omega_{ij}\left(x_{j}-\bar{X}\right)$$
where $S_{i}^{2}$ is the variance of attribute value *x_i_*. The local spatial autocorrelation results show four types, including high-high, low-low, high-low, and low-high.

#### 3.2.5. Model Construction and Selection of Influencing Factors

The coupling coordination degree of PLES index presents significant spatial heterogeneity in space. In this study, Ordinary least squares (OLS) and GWR model were used to measure the influencing factors of coupling coordination degree of PLES at county scale in China [45,46]. Specifically, OLS model analyzes the influencing factors of the coupling coordination degree of PLES in China from a global perspective without considering the influence of spatial factors. The GWR model considers the spatial heterogeneity of influencing factors on the coupling coordination degree of PLES in China. The calculation equation of OLS model is as follows:(11)$$y_{i}=\beta_{0}+\sum_{j=1}^{k}\beta_{j}x_{ij}+\varepsilon_{i}\;(i=1,\;2,\;3...,\;m;\;j=1,\;2,\;3,...,\;n)$$
where *β_0_* is the intercept; *β_i_* is the regression coefficient and *ε_i_* is the random error term.

GWR extends OLS to describe spatial non-stationary by incorporating spatial dimensions into modeling. GWR model can generate local rather than global estimates and help visualize spatial non-stationary relationships [47]. The method for determining local estimates relies on the Gaussian distance decay function, where the contribution of a sample is based on the weight of its proximity to the position of the sample under consideration [47]. The calculation equation of GWR model is as follows:(12)$$y_{i}=\beta_{0}\left(m_{i},n_{i}\right)+\sum_{j=1}^{k}\beta_{j}\left(m_{i},n_{i}\right)x_{ij}+\varepsilon_{i}$$
where (*m_i_*, *n_i_*) is the geospatial coordinate of sample point *i*. *β_j_* (*m_i_*, *n_i_*) is the regression coefficient *j* of sample point *i*. The positive and negative values of *β_j_* (*m_i_*, *n_i_*) indicate the promoting or inhibiting effect of *x_ij_* on *y_i_*.

Territorial spatial pattern is influenced by natural environment factors, socioeconomic factors, and landscape pattern characteristics [1,33]. Based on previous literature, this study constructed the influencing factors of coupling coordination degree of PLES index in China from three dimensions of natural environment, social economy, and landscape pattern (Table 3). The dimension of natural environment includes altitude and precipitation to represent terrain and climate factor, respectively [1,35]. Socioeconomic dimensions include land use intensity and population density, respectively, representing human activity intensity and population pressure [1,48]. Landscape pattern dimensions include patch density and interspersion and juxtaposition index, which respectively represent the characteristics of landscape fragmentation and agglomeration [33,49].

## 4. Results

### 4.1. Spatio-Temporal Distribution of PLES Index in China from 2000 to 2020

On the whole, the PSI and LSI in China showed a continuous growth trend, while the ESI showed a continuous decline. As shown in Figure 1, Figure 2 and Figure 3, the proportion of counties with PSI growth was 52.99% and 71.24% during 2000–2010 and 2010–2020, respectively. The proportion of LSI was 76.29% and 85.53%, respectively, while the proportion of ESI was only 22.52% and 26.62%, respectively. In 2000, 2010, and 2020, China’s PSI was 1.652, 1.662, and 1.706, LSI was 0.414, 0.477, and 0.598, ESI was 3.536, 3.488, and 3.398, respectively. The ESI of China was evidently higher than the PSI, and the LSI was the lowest (Appendix A). Moreover, from the perspective of spatial distribution, PSI and LSI maintained a high degree of spatial coincidence, mainly distributed in the east of Hu Line. Specifically, the counties with a PSI above 2.5 were mainly distributed in northeast China, Sichuan Basin, North China Plain, Middle-Lower Yangtze Plain, and Southeast Hilly region. The spatial agglomeration of living space was particularly obvious, which was mainly distributed in the North China Plain and the Northeast China Plain, the Kuan-chung Plain, the main urban agglomerations (e.g., Tianjin-Beijing-Hebei urban agglomeration, Bohai Rim Economic Zone, the Middle Reaches of the Yangtze River urban agglomeration, Yangtze River Delta urban agglomeration, the Pearl River delta urban agglomerations), and the capital cities and surrounding areas. Forestland, grassland, cultivated land, water area, and wetland are the main ecological spaces with a wide distribution range. The ecological space was mainly distributed in the Greater Khingan Mountains, Lesser Khingan Mountains, Changbai Mountains, and T’ai-hang Mountains, and the ecological space in the Kunlun Mountains, Bayankla Mountains, Qinling Mountains, while the southern Dabie Mountains was significantly higher than that in the north. 

### 4.2. Gravity Centers of PLES Index in China from 2000 to 2020

Based on the gravity centers model, this study calculated the evolution law of gravity centers of the PLES index in 2000, 2010, and 2020, and generally found that the living space, production space, and ecological space had a trend of migration to western China during the study period (Figure 4). Specifically, the living space moved 34.241 km to southwest China from 2000 to 2010, and 28.382 km to northwest China from 2010 to 2020. Production space continued to migrate to northwest China during the study period, but the migration was not evident, with a migration of 1.204 km to southwest China from 2000 to 2010, and 9.257 km to northwest China from 2010 to 2020. The ecological space shifted 6.243 km to northwest China from 2000 to 2010, and 3.858 km to southeast China from 2010 to 2020.

### 4.3. Coupling Coordination Degree of PLES Index in China from 2000 to 2020

#### 4.3.1. Coupling Coordination Degree of Subsystems of PLES Index in China from 2000 to 2020

It can be seen in Figure 5 that the moderately unbalanced category was concentrated in the northeast and southeast of China, the seriously unbalanced category was mainly distributed in the west of Hu Line, while the basically balanced category was mostly situated in North China Plain and Middle-Lower Yangtze Plain. In 2000, 2010, and 2020, the coupling coordination degree of PSI and LSI in China are 0.330, 0.341, and 0.374, respectively, in the moderately unbalanced category. Specifically, the number of units in the moderately unbalanced category was the largest, accounting for 36.778%, 36.357%, and 37.793% in 2000, 2010, and 2020, respectively, decreased first and then increased. The second was the seriously unbalanced category, accounting for 28.827%, 27.636%, and 21.611%, respectively, showing a trend of continuous decrease. Moreover, the proportion of basically balanced units is 26.445%, 25.954%, and 26.410%, decreased first and then increased, followed by moderately balanced units, accounting for 5.289%, 7.040%, and 9.947%, respectively, showing a continuous increase. During the study period, the number of highly balanced units accounted for the lowest proportion (2.662%, 3.012%, and 4.238%, respectively), also showed a continuously increasing trend.

In general, the moderately balanced category is mainly distributed in the east of Hu Line, while in the west of Hu Line are mainly seriously unbalanced, moderately unbalanced, and basically balanced categories. As shown in Figure 6, the coupling coordination degree of PSI and ESI in China in 2000, 2010, and 2020 are 0.669, 0.660, and 0.664, respectively, in the moderately balanced category. The number of units in the moderately balanced category is the largest, accounting for 84.658%, 83.713%, and 83.853% in 2000, 2010, and 2020, respectively, showing a decreasing first and then increasing trend, followed by the basically balanced category, accounting for 8.932%, 9.772%, and 9.737%, respectively, increased first and then decreased. In addition, the proportion of moderately unbalanced category units is 4.343%, 4.378%, and 4.203%, respectively, also showing an increasing first and then decreasing trend. Furthermore, the number of seriously unbalanced category units accounted for 2.067%, 2.137%, and 2.207%, respectively, showing a continuously increasing trend. It is worth noting that there were no highly balanced category units in China during the study period. 

On the whole, the seriously unbalanced category was concentrated in the west of Hu Line. Units with moderately unbalanced and basically balanced categories were mainly distributed in the east of Hu Line. Among them, the basically balanced category was mostly situated in North China Plain and Middle-Lower Yangtze Plain. As shown in Figure 7, in 2000, 2010, and 2020, the coupling coordination degree of LSI and ESI in China are 0.360, 0.370, and 0.398, respectively, in the moderately unbalanced category. First, the number of units in the moderately unbalanced category is the largest, accounting for 42.627%, 40.771%, and 40.105% in 2000, 2010, and 2020, respectively, showing a decreasing trend. Moreover, the number of basically balanced category units is 36.918%, 38.669%, and 41.226%, respectively, showing a continuous increase, followed by the seriously unbalanced category, accounting for 15.116%, 14.256%, and 9.527%, respectively, showing a continuous decreasing trend. In addition, the proportion of moderately balanced category units was 5.289%, 6.305%, and 9.142%, respectively, showing a significant increase. Similarly, there were no highly balanced category units in China during the study period.

#### 4.3.2. Coupling Coordination Degree of PLES Index in China from 2000 to 2020

In 2000, 2010, and 2020, the coupling coordination degree of the PLES index in China was 0.417, 0.423, and 0.449, respectively, in the basically balanced category. In 2000, 2010, and 2020, the number of units in the basically balanced category was the largest, accounting for 44.133%, 43.713%, and 43.117%, respectively, showing a continuous decreasing trend. The second was the moderately unbalanced category, accounting for 35.832%, 34.711%, and 29.282%, respectively, showing a trend of continuous decrease. The proportion of moderately unbalanced units was 11.384%, 13.275%, and 20.245%, showing a significant increasing trend. During the study period, the number of seriously unbalanced units accounted for the lowest proportion (8.651%, 8.301%, and 7.356%, respectively), showing a continuous decreasing trend. It can be seen that the basically balanced category was the main type of coupling coordination, followed by the moderately unbalanced category, while the seriously unbalanced category had the least number of county units.

From the perspective of spatial distribution (Figure 8, Figure 9, Figure 10, Figure 11 and Figure 12), the west of Hu Line was mainly characterized by the seriously unbalanced-LSI lag subcategory, seriously unbalanced-PSI lag subcategory, and moderately unbalanced-LSI lag subcategory. The eastern counties of Hu Line were mainly the moderately unbalanced category and above. Among them, the moderately unbalanced category was mainly distributed in Wushan, Lingnan, Yunnan-Kweichow Plateau, and Wuyi Mountains in the south of the Qinling Mountains and The Greater and Lesser Khingan Mountains, and Changbai Mountains in the northeast. The basically balanced category was mainly distributed in three plain regions of China (e.g., Northeast China Plain, North China Plain, Middle-Lower Yangtze Plain), Sichuan Basin, and Pearl River Delta whereas the basically balanced-LSI lag subcategory was the main subcategory, with the trend of expanding. The moderately balanced category was mainly distributed in the North China Plain, Sichuan Basin, Pearl River Delta, and some provincial cities, with the moderately balanced-ESI lag category the main subcategory.

### 4.4. Spatial Autocorrelation Analysis of Coupling Coordination Degree of PLES in China from 2000 to 2020

The results of spatial autocorrelation analysis showed that the global Moran’ I of the coupling coordination degree of PLES in China in 2000, 2010, and 2020 were 0.788, 0.790, and 0.769, respectively, with significance at the 0.001 level and Z-values of 68.139, 67.487, and 66.416, respectively. It showed that the coupling coordination degree of PLES in China had significant spatial agglomeration, and the spatial agglomeration situation increased first and then decreased. As shown in Figure 13, the low-low type was the main relationship type, widely distributed in western China, followed by the high-high type, mainly situated in North China Plain, Yangtze River Delta, Pearl River Delta, Jianghan Plain, Chengdu Plain, Northeast China Plain, and some provincial capital cities (e. g., Changsha, Fuzhou, Xi’an, Taiyuan). Low-high and high-low type areas were rare and sporadically distributed throughout the country.

### 4.5. Spatial Heterogeneity Analysis of Influencing Factors of Coupling Coordination Degree of PLES in China from 2000 to 2020

According to the regression statistical results of OLS and GWR model, GWR has better goodness of fit (Table 4 and Table 5). Moreover, results showed that the ratio of positive coefficients of land use intensity, patch density, and interspersion and juxtaposition index was significantly higher than that of negative coefficients (Table 6). The coefficient positive ratio of land use intensity reached 100%, while the coefficient negative ratio of population density and altitude both exceeded 75%. The results showed that the effects of these factors on the coupling coordination degree of PLES had no significant spatial heterogeneity. The positive and negative coefficients of precipitation were balanced in proportion, indicating that the effect of precipitation factors on the coupling coordination degree of PLES had significant spatial difference.

The spatial distribution of the regression coefficients of the influencing factors of the coupling coordination degree of PLES index in China in 2000, 2010, and 2020 was exhibited in Figure 14, Figure 15 and Figure 16. Specifically, the land use intensity positively impacted the degree of coupling coordination of the PLES index in China, which increased gradually from eastern China to western China. Population density negatively impacted the coupling coordination degree of the PLES index in China, especially in the northeast of China and Tibet. Generally, the increase in population density will lead to a simultaneous increase in the demand for PLES, which will lead to the disordered development of space, the excessive utilization of resources, the intensification of PLES conflict, and a negative impact on the coupling coordination degree of PLES index.

In terms of altitude factors, generally, the increase in altitude would limit the improvement of the coupling coordination degree of the PLES index, especially in the southeast of China, where only a few counties are located in Inner Mongolia and Heilongjiang province. Precipitation had positive and negative effects on the coupling coordination degree of the PLES index. The positive effects were mainly distributed in northeast China, northwest China, and southern China, while in other regions they were negative. Landscape fragmentation can effectively promote the coupling coordination degree of the PLES index. In northeast China and most of western China, landscape fragmentation can significantly improve the coupling coordination degree of the PLES index. In northeast China, the spatial coupling coordination degree of PLES index factors was negatively affected by the interspersion and juxtaposition index, and the coupling coordination degree of PLES index factors was significantly improved by the interspersion and juxtaposition indexes in other regions, especially in Southern China. The interspersion and juxtaposition index represents the degree of landscape agglomeration, and the larger the value is, the more landscape elements are adjacent to other landscape elements. The higher the interspersion and juxtaposition index, the higher the probability of increasing the proximity between patches of various types of land space, and the higher the coupling coordination degree of the PLES index.

## 5. Discussion

### 5.1. Interpretation of Results

#### 5.1.1. Spatio-Temporal Distribution of PLES Index in China from 2000 to 2020

In general, during the study period, the PSI and LSI in China showed a continuous growth, while the ESI showed a continuous decline. Kong et al. also obtained the same research results: regarding the spatio-temporal evolution of the PLES, China’s production space and living space increased while ecological space decreased from 1990 to 2018 [35]. Moreover, PLES showed different spatial distribution characteristics, which was basically consistent with the research of Liu et al. [29]. The counties with a PSI above 2.5 were mainly distributed in northeast China, Sichuan Basin, North China Plain, Middle-Lower Yangtze Plain, and Southeast Hilly region. This is mainly due to the flat terrain and suitable climate in this part of the country, which is the main food production base with large areas of cultivated land and construction land, providing sufficient space for production. Additionally, the spatial agglomeration of living space was particularly obvious (Figure 2), mainly distributed in the North China Plain and the Northeast China Plain, the Kuan-chung Plain, the main urban agglomerations, and the capital cities and surrounding areas. It may be because this part of the region has a high level of economic development with frequent human activities, and it is the main place where the main social and economic activities take place, gathering the main distribution of construction land and is the main living space. In addition, ecological space was mainly distributed in the Greater Khingan Mountains, Lesser Khingan Mountains, Changbai Mountains, and T’ai-hang Mountains. These regions are key ecological function zones, and high-intensity human activities such as rapid urbanization and land resource development are greatly restricted (Figure 3). Therefore, ESI of these areas is high.

#### 5.1.2. Gravity Centers of PLES Index in China from 2000 to 2020

The gravity centers model displayed that China’s PLES generally transferred to western China, with a large transfer extent of living space and a relatively small transfer extent of production space and ecological space. The possible reason is that the implementation of the western development policy since 2000 has greatly improved the level and speed of economic development in northwest China, and the massive input of human, material, and financial resources has dramatically improved the production, living, and ecological conditions in northwest China. However, the great development of western China mainly promotes economic growth through a large amount of physical capital support, especially infrastructure investment, making the transfer range of living space significantly larger than production space and ecological space during the study period.

#### 5.1.3. Coupling Coordination Degree of PLES Index in China from 2000 to 2020

The west of Hu Line is mainly characterized by the seriously unbalanced subcategory. The main reason is that the counties to the west of the Hu Line are dominated by plateau and desert, which are cut across by major mountains, and the terrain is raised and divided into different climatic zones. The natural ecology is fragile, the unused land is widely distributed, the living and production space is rare and unevenly distributed, and the population and economic carrying capacity are low. However, in recent years, with the development of western China, the shifting momentum from the seriously unbalanced to the moderately unbalanced category is obvious. The eastern counties of Hu Line are mainly the moderately unbalanced category and above. Among them, the moderately unbalanced category is mainly distributed in Wushan, Lingnan, Yunnan-Kweichow Plateau, and Wuyi Mountains in the south of the Qinling Mountains and The Greater and Lesser Khingan Mountains, and in the northeast of the Changbai Mountains. The main reason is that these regions are predominantly mountains and hills, rich in forest resources, and abundant in ecological space and production space. However, their living conditions are greatly affected by topographic factors, and the construction land is relatively small, so the living space is narrow. The basically balanced category is mainly distributed in three plain regions of China (e.g., Northeast China Plain, North China Plain, Middle-Lower Yangtze Plain), Sichuan Basin, and Pearl River Delta. The main reason is that these regions have broad plains, abundant cultivated land and grassland resources, and extensive production and ecological space. However, these regions are mainly agricultural production regions with important grain production bases, and their construction and development activities are limited to a certain extent, resulting in lagging development of living space. Moreover, the superior geographical conditions and natural resource endowment also make the population of these regions gather continuously in recent years, the construction land is expanding continuously, and the basically balanced category is expanding continuously. The moderately balanced category is mainly distributed in the North China Plain, Sichuan Basin, Pearl River Delta, and some provincial cities. Most of these regions are optimized development zones and key development zones with clear main functions, frequent economic activities, high population density, and high demand for living and production space, making ecological space seriously occupied. With the continuous advancement of urbanization, the trend of this category of expansion to surrounding areas is increasingly significant. Overall, in the plain regions, peripheral areas of metropolises and urban agglomerations, the coupling coordination degree of the PLES index is evidently higher than those of other regions. It may be because these regions are major production and living places, and a better regional system of man-land relationship has been formed. There is a broad ecological space in mountainous and hilly areas with high altitudes, while the coupling coordination degree of PLES in these regions is obviously low due to the terrain and climate factors.

#### 5.1.4. Spatial Autocorrelation Analysis of Coupling Coordination Degree of PLES in China from 2000 to 2020

Spatial autocorrelation analysis shows that low-low and high-high types are the main relationship types. Previous studies have also shown that the spatial distribution of the PLES presented a strong positive spatial autocorrelation [51]. The agglomeration trend of cities with high-high and low-low types was obvious, while cities with high-low and low-high types were scattered under the influence of “siphon”. In addition, we find that the low-low type is widely distributed in western China, while the high-high type is mainly situated in North China Plain, Yangtze River Delta, Pearl River Delta, Jianghan Plain, Chengdu Plain, and Northeast China Plain. Through a series of measures, for example, the great western development, the resource utilization efficiency of the western regions has been improving, and in some aspects, the extent of improvement is greater than the national average. However, due to the backward economic development, poor natural conditions, and fragile ecological environment, the coupling coordination of PLES is still an important challenge in western China [52]. By contrast, benefiting from better development conditions, there is a higher coupling coordination degree of PLES in regions such as North China Plain, Yangtze River Delta, and Pearl River Delta.

#### 5.1.5. Spatial Heterogeneity Analysis of Influencing Factors of Coupling Coordination Degree of PLES in China from 2000 to 2020

The land use intensity positively impacts the degree of coupling coordination of the PLES index in China and the positive impact increases gradually from eastern China to western China. Land use intensity can effectively characterize the intensity of human activities. The improvement of land use intensity means the intensification of human transformation of nature. In recent years, especially in western China, with the implementation of the great western development strategy in China, the intensity of human activities has been increasing. Consequently, the increase of artificial and semi-artificial landscape in optimized development zones and key development zones effectively increase production and living space, thus improving the coupling coordination degree of PLES and promoting the coordinated development of regional PLES. Population density negatively impacts the coupling coordination degree of the PLES index in China, especially in northeast China and Tibet. In general, the increase in population density will lead to a simultaneous increase in the demand for PLES, which will produce the disordered development of space, the excessive utilization of resources, the intensification of PLES conflict, and a negative impact on the coupling coordination degree of the PLES index.

In terms of altitude factors, generally, the increase in altitude will limit the improvement of the coupling coordination degree of the PLES index, especially in southeast China. Altitude has a strong restriction on human activities, which results in the limitation of the co-development of PLES with the increase of altitude. This is also why the production space and living space in plain areas are significantly higher than the ecological space. Precipitation has different effects on the coupling coordination degree of the PLES index. The positive effects are mainly distributed in northeast China, northwest China, and south China, while the other regions are negative. It also indirectly reflects the differences in precipitation in China. Landscape fragmentation positively affects the coupling coordination degree of the PLES index in northeast China and most of western China. This may be related to the land use intensity, where in the higher the intensity of land use, the higher the degree of landscape fragmentation. Therefore, as land use intensifies, landscape fragmentation positively affects the coupling coordination degree of the PLES index. Similarly, the higher the interspersion and juxtaposition index, the higher the landscape fragmentation. Therefore, in northeast China, areas with less human activities, the spatial coupling coordination degree of the PLES index factors is negatively affected by the interspersion and juxtaposition index. While in regions, for example, South China, with intense human activities, the coupling coordination degree of PLES index factors is significantly improved by the interspersion and juxtaposition index.

### 5.2. Policy Implications

This study established a classification and evaluation system for PLES and analyzed the spatio-temporal characteristics, coupling coordination degree, and influencing factors of PLES in China from 2000 to 2020. The results can offer a scientific reference for the optimization of territorial space layout and the promotion of high-quality development of territorial space. The following policy suggestions are proposed based on these findings:

First, on the whole, to improve the quality of coupled and coordinated utilization of PLES, it is necessary to change the development mode of territorial space that is oriented by efficient utilization and constrained by environmental protection, and form the concept of improving the quality of urbanization development with “people” as the core. It is suggested that the “three-line” management and control of the PLES layout should be optimized to build a high-quality territorial spatial development pattern. Based on the zoning of the main functional zones, the types of main functional zones should be defined in each region to promote high-level coupling and coordination of PLES [53].

Specifically, we should pay more attention to the unbalanced regions of PLES, especially seriously unbalanced regions. Moreover, for different subcategories of seriously unbalanced, formulating differentiated measures is necessary. In areas with a seriously unbalanced-LSI lag subcategory, the urgent task is to improve infrastructure and social security, improving the level of living space. Additionally, in regions with a seriously unbalanced-PSI lag subcategory, focus should be given to deepening the reform of the economic system and promoting the optimization of production space. Whereas in regions with a seriously unbalanced-ESI lag subcategory, we should give priority to the construction of ecological civilization and give consideration to economic development and ecological protection [51].

Moreover, we should consider the differences in influencing factors of the coupling coordination degree of PLES. As the research results show, socioeconomic factors and landscape pattern can effectively promote the coupling coordination degree of PLES. Therefore, we should pay attention to the regions with backward economic development to promote their high-quality socio-economic development to further drive and optimize the coupling coordination degree of PLES. Meanwhile, in areas with low-developed land, properly increasing land use intensity and landscape fragmentation will also help to integrate the PLES and improve the coupling coordination degree of PLES.

### 5.3. Limitation and Future Direction

The coupling coordination level of PLES is affected by nature, social economy, and landscape pattern to different degrees [1,2]. To reveal the internal mechanism of the coupling coordination degree of PLES, it is necessary to deepen the research on the formation mechanism and evolution law of regional function. Besides, it is highly recommended to go for further research on the optimal allocation theory and technology of PLES and focus on the multi-function theory of land use and the theory of man-land relationship based on scientifically defining the concept connotation of PLES [54]. The factors of regional socioeconomic development, resource and environmental management, and territorial space should be holistically taken into account. Exploring the internal relationship among PLES, and clarifying the mechanisms of different influencing factors to promote scientific policy-making and targeted implementation should be considered [2,17]. The study on the coupling coordination of PLES can clarify the optimization direction of the coordinated development of PLES. However, there may be some differences in the coupling coordination of PLES at different spatial scales. The coupling coordination analysis at provincial, prefectural, county, township, and even patch levels can provide different reference information for spatial optimization, and the study on coupling coordination of PLES at multiple scales is of great importance. In addition, it is necessary to form a “quality” and “quantity” view of the coupling coordination of PLES, attach importance to the spatial “quality” and “quantity” characteristics of PLES identification and optimization, and deeply analyze the difference of spatial coupling coordination “quality” based on the problem of coupling coordination “quantity” of the PLES.

## 6. Conclusions

Based on the remote sensing data of land use in China from 2000 to 2020, this study constructed a land use classification system for PLES, measured the spatio-temporal patterns of PSI, ESI, and LSI at a county scale in China from 2000 to 2020, and measured the spatio-temporal differentiation characteristics of PLES in China by using the gravity center analysis method. The coupling coordination degrees of PSI, ESI, and LSI were measured by the coupling coordination model, and the spatial heterogeneity of influencing factors of PLES index coupling coordination degree was explored by the GWR model. The results are as follows:

(1) During the study period, the PSI and LSI in China increased continuously, while the ESI decreased continuously. The ESI of China was significantly higher than the PSI, and the LSI was the lowest. PSI and LSI maintained a high degree of spatial coincidence, mainly distributed in the plain areas to the east of Hu Line, urban agglomerations, and the surrounding areas of big cities. The ESI in the mountainous areas was significantly higher than that in plain areas. During the study period, the LSI, PSI, and ESI all tended to migrate to the western region of China, and the migration range of living space was larger than that of production space and ecological space.

(2) During the study period, the basically balanced category was the main type of coupling coordination, followed by the moderately unbalanced category, while the seriously unbalanced category had the least number of county units. Among them, the number of units of the basically balanced category-LSI lag subcategory, and the moderately unbalanced-LSI lag subcategory accounted for more than 29% and 41%, respectively, which were the main coupling coordination subcategories. In terms of spatial distribution, the west of Hu Line was mainly characterized by the seriously unbalanced category, while the east of Hu Line was mainly characterized by the moderately unbalanced category and above. Moreover, the spatial autocorrelation of the coupling coordination degree of PLES was high-high and low-low types.

(3) According to the GWR regression results, natural factors were the main reason limiting the coordinated development of the PLES index. Socioeconomic factors could effectively promote the coupling coordinated development of PLES, while landscape pattern also positively affected the coupling coordination of PLES.

## Figures and Tables

**Figure 1 ijerph-19-10370-f001:**
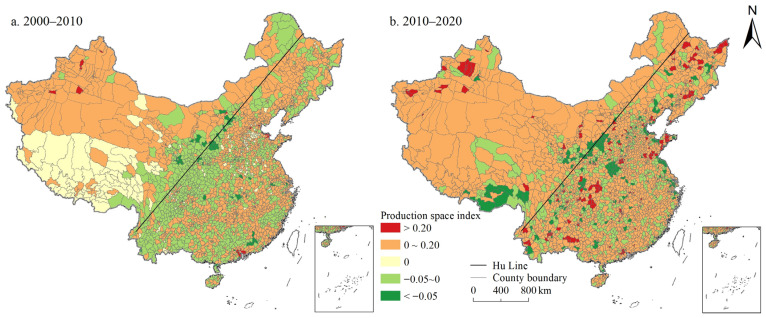
Spatial distribution of PSI changes in China during 2000–2020.

**Figure 2 ijerph-19-10370-f002:**
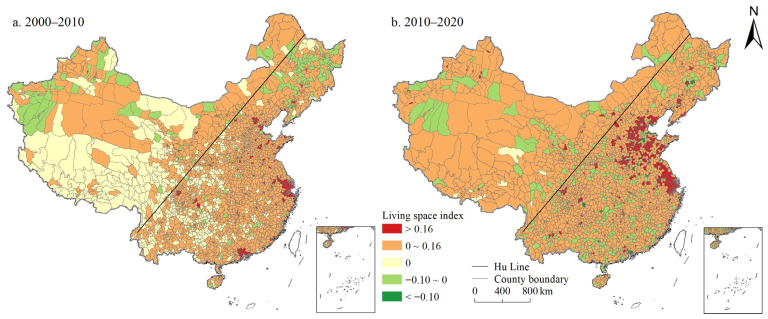
Spatial distribution of LSI changes in China during 2000–2020.

**Figure 3 ijerph-19-10370-f003:**
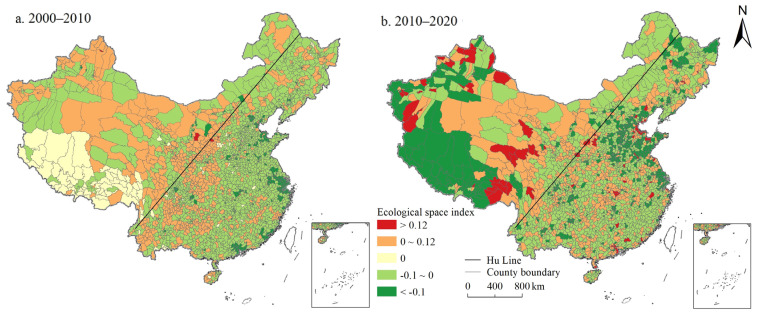
Spatial distribution of ESI changes in China during 2000–2020.

**Figure 4 ijerph-19-10370-f004:**
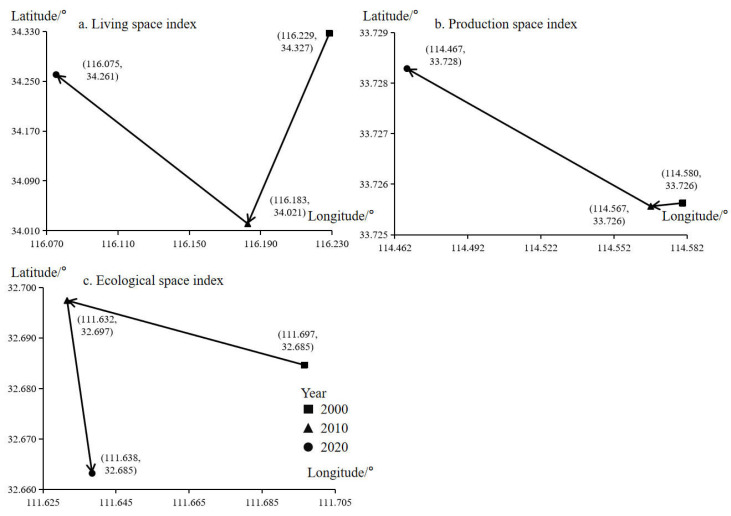
Evolution trajectory of PLES index gravity centers in China during 2000–2020.

**Figure 5 ijerph-19-10370-f005:**
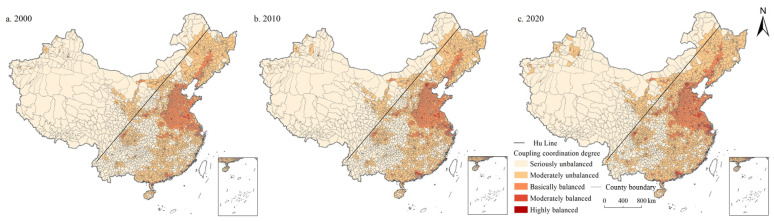
Spatial distribution of coordination degree of PSI and LSI in China during 2000–2020.

**Figure 6 ijerph-19-10370-f006:**
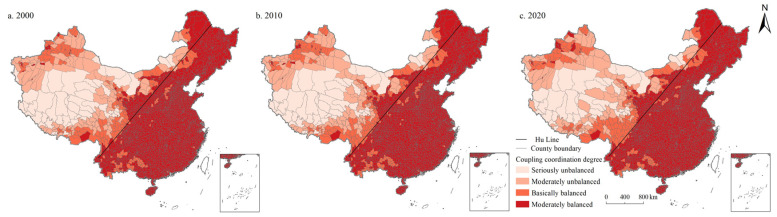
Spatial distribution of coordination degree of PSI and ESI in China during 2000–2020.

**Figure 7 ijerph-19-10370-f007:**
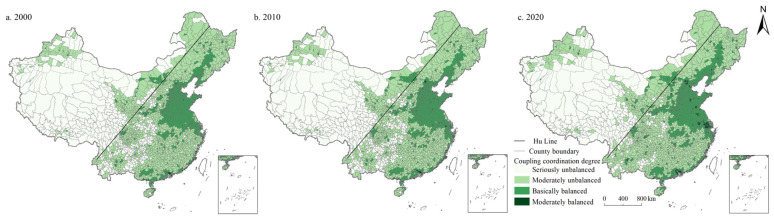
Spatial distribution of coordination degree of LSI and ESI in China during 2000–2020.

**Figure 8 ijerph-19-10370-f008:**
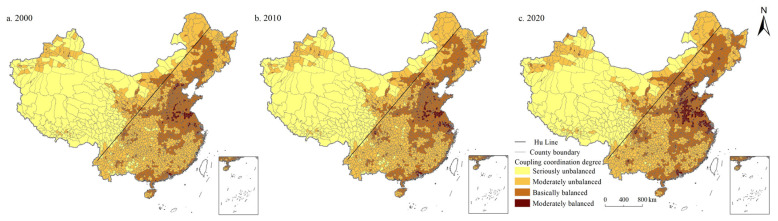
Spatial distribution of coordination degree of PLES in China during 2000–2020.

**Figure 9 ijerph-19-10370-f009:**
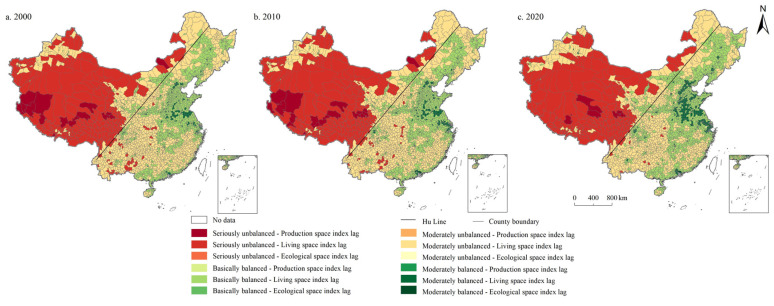
Spatial distribution of coordination degree of PLES index subcategory in China during 2000–2020.

**Figure 10 ijerph-19-10370-f010:**
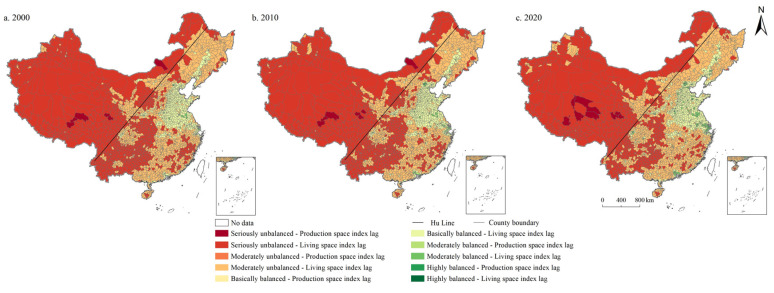
Spatial distribution of subcategories of coordination degree of PSI and LSI in China during 2000–2020.

**Figure 11 ijerph-19-10370-f011:**
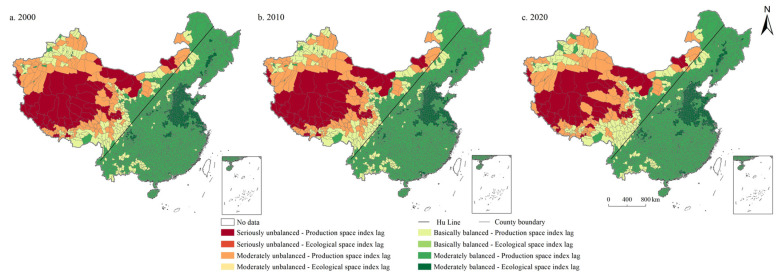
Spatial distribution of subcategories of coordination degree of PSI and ESI in China during 2000–2020.

**Figure 12 ijerph-19-10370-f012:**
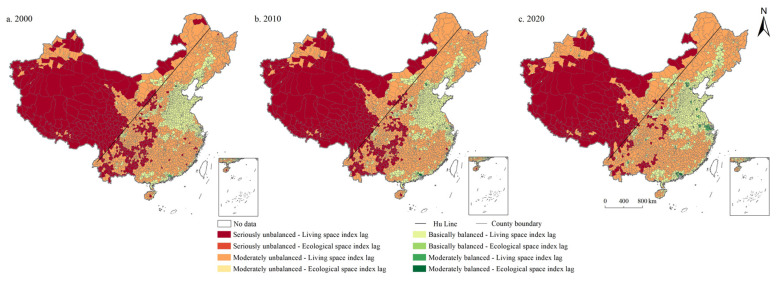
Spatial distribution of subcategories of coordination degree of LSI and ESI in China during 2000–2020.

**Figure 13 ijerph-19-10370-f013:**
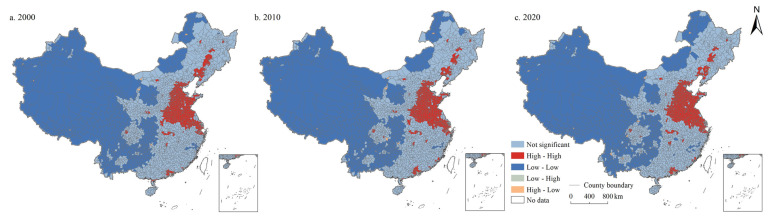
Spatial autocorrelation of coordination degree of production-living-ecological space in China during 2000–2020.

**Figure 14 ijerph-19-10370-f014:**
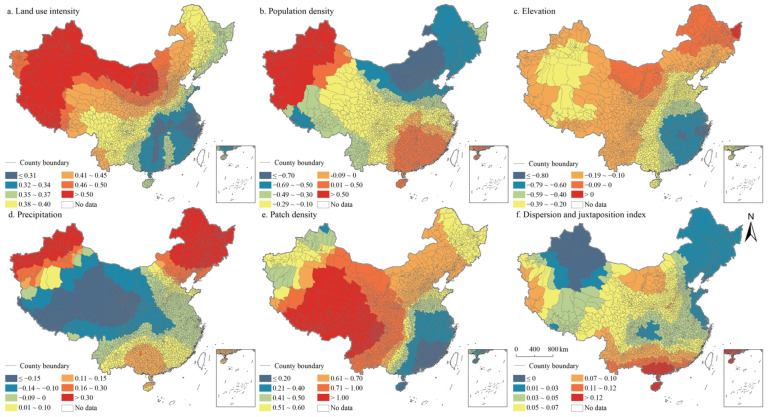
Spatial distributions of the regression coefficients of GWR models in China in 2000.

**Figure 15 ijerph-19-10370-f015:**
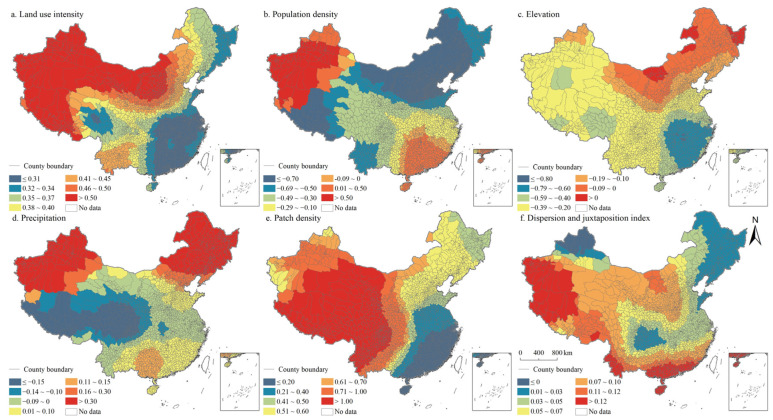
Spatial distributions of the regression coefficients of GWR models in China in 2010.

**Figure 16 ijerph-19-10370-f016:**
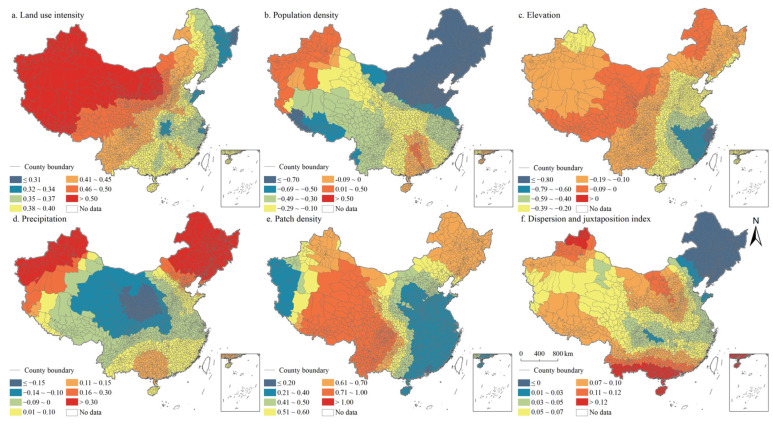
Spatial distributions of the regression coefficients of GWR model in China in 2020.

**Table 1 ijerph-19-10370-t001:** Land use classification and assessment of PLES.

First-Level Land Use Types	Code	Second-Level Land Use Types	Production Land	Living Land	Ecological Land	First-Level Land Use Types	Code	Second-Level Land Use Types	Production Land	Living Land	Ecological Land
Cultivated land	11	Paddy field	3	0	3	Construction land	52	Rural settlements	3	5	0
	12	Dry land	3	0	3		53	Other construction land	5	1	0
Forestland	21	Forest	1	0	5	Unused land	61	Sandy land	0	0	1
	22	Shrub	0	0	5		62	Gobi	0	0	1
	23	Woods	0	0	5		63	Salina	0	0	1
	24	Others	0	0	5		65	Bare soil	0	0	1
Grassland	31	Dense grassland	0	0	5		66	Bare rock	0	0	1
	32	Moderate grassland	0	0	5		67	Others	0	0	1
	33	Sparse grassland	0	0	3	Wetland	44	Permanent ice andsnow	0	0	5
Water area	41	Stream and rivers	0	0	5		45	Beach and shore	0	0	5
	42	Lakes	0	0	5		46	Bottomland	0	0	5
	43	Reservoir and ponds	1	0	1		64	Swampland	0	0	5
	51	Urban built-up	3	5	0						

**Table 2 ijerph-19-10370-t002:** Classification of coupling coordination degree of PLES.

Category	Coupling Coordination Degree	Sub-Category	Relative Relation
Seriously unbalanced	0 ≤ D ≤ 0.2	Seriously unbalanced PSI lag	*N* = PSI
		Seriously unbalanced LSI lag	*N* = LSI
		Seriously unbalanced ESI lag	*N* = ESI
Moderately unbalanced	0.2 ≤ D ≤ 0.4	Moderately unbalanced PSI lag	*N* = PSI
		Moderately unbalanced LSI lag	*N* = LSI
		Moderately unbalanced ESI lag	*N* = ESI
Basically balanced	0.4 ≤ D ≤ 0.6	Basically balanced PSI lag	*N* = PSI
		Basically balanced LSI lag	*N* = LSI
		Basically balanced ESI lag	*N* = ESI
Moderately balanced	0.6 ≤ D ≤ 0.8	Moderately balanced PSI lag	*N* = PSI
		Moderately balanced LSI lag	*N* = LSI
		Moderately balanced ESI lag	*N* = ESI
Highly balanced	0.8 ≤ D ≤ 1	Highly balanced PSI lag	*N* = PSI
		Highly balanced LSI lag	*N* = LSI
		Highly balanced ESI lag	*N* = ESI

Notes: *N* = min {PSI, LSI, ESI}.

**Table 3 ijerph-19-10370-t003:** Factors influencing the coupling coordination degree of PLES in China.

Dimensions	Metrics	Calculation Method
Natural environment	DEM	DEM treated by depression filling was extracted in ArcGIS 10.3 using the Zonal Statistic tool.
	Precipitation	It was extracted by Arc Toolbox/Spatial Analyst Tools/Zonal/Zonal Statistics tool in ArcGIS 10.3.
Social economy	Land use intensity	Referring to the measurement model of land use degree proposed by Zhuang et al. (1997), the natural balance retention state of China’s land system under the effect of social factors was measured [50].
	Population density	It was extracted by Arc Toolbox/Spatial Analyst Tools/Zonal/Zonal Statistics tool in ArcGIS 10.3.
Landscape pattern	Patch density	$PD=\sum_{i=1}^{m}N_{i}/LA$ where *PD* is the patch density. m is the total number of landscape types. *LA* is the total landscape area of the study area. *N_i_* is the number of patches of land use type *i*.
	Interspersion and juxtaposition index	$IJI=\frac{-\sum_{k=1}^{m}\left[\frac{e_{ik}}{\sum_{k=1}^{m}e_{ik}}\ln\left(\frac{e_{ik}}{\sum_{k=1}^{m}e_{ik}}\right)\right]}{\ln(m-1)}\times 100\%$ where *IJI* is the interspersion and juxtaposition index; *e_ik_* is the total edge length between the patch types *i* and *k* in the landscape. *m* is the number of patch types.

**Table 4 ijerph-19-10370-t004:** Regression results of OLS model during 2000–2020.

Variable	2000					2010					2020				
	Coefficient	StdError	t-Statistic	Probability	VIF	Coefficient	StdError	t-Statistic	Probability	VIF	Coefficient	StdError	t-Statistic	Probability	VIF
Intercept	0.354	0.008	25.481	0.000		0.538	0.011	49.512	0.000		0.193	0.008	23.325	0.000	
Land use intensity	0.411	0.010	43.179	0.000	2.291	0.371	0.013	28.941	0.000	2.329	0.445	0.009	46.931	0.000	2.407
Population density	−0.277	0.026	−10.626	0.000	1.447	−0.391	0.033	−11.903	0.000	1.453	−0.522	0.023	−22.943	0.000	1.442
Altitude	−0.246	0.010	−24.722	0.000	2.040	−0.402	0.014	−29.615	0.000	2.059	−0.188	0.010	−18.167	0.000	2.124
Precipitation	−0.014	0.008	−1.607	0.108	1.433	−0.002	0.011	−0.149	0.882	1.440	0.007	0.008	0.835	0.404	1.452
Patch density	0.558	0.026	21.256	0.000	1.425	0.526	0.036	14.729	0.000	1.403	0.501	0.023	21.672	0.000	1.355
Interspersion and juxtaposition in-dex	0.054	0.007	7.619	0.000	1.247	0.073	0.010	7.522	0.000	1.271	0.057	0.007	7.735	0.000	1.259
R^2^		0.778					0.705					0.761			
R^2^ Adjust		0.777					0.705					0.761			
F-statistics		1657.816					1134.422					1509.371			
AICc		−6656.008					−4912.572					−6537.363			

**Table 5 ijerph-19-10370-t005:** Test result of the GWR model.

	2000	2010	2020
Bandwidth	768,294.568	696,442.707076	798,886.278
AICc	−7405.841	−5678.214	−7423.368
R^2^	0.834	0.782	0.829
R^2^ Adjust	0.830	0.777	0.826

**Table 6 ijerph-19-10370-t006:** Statistical results of the GWR model regression coefficient during 2000—2020.

	Variable	Land Use Intensity	Population Density	Altitude	Precipitation	Patch Density	Interspersion and Juxtaposition Index	Intercept	*N*
2000	Min	0.262	−0.798	−0.852	−0.234	0.054	−0.089	0.073	2849
	Upper-quartile	0.334	−0.544	−0.526	−0.093	0.360	0.030	0.141	
	Median	0.377	−0.244	−0.253	−0.015	0.572	0.049	0.175	
	Lower-quartile	0.424	−0.037	−0.174	0.107	0.721	0.067	0.265	
	Max	0.715	3.520	0.152	0.547	1.356	0.186	0.381	
	Positive (%)	100	21.80	0.28	45.45	100	97.33	100	
	Negative (%)	0	78.20	99.72	54.53	0	2.67	0	
2010	Min	0.239	−4.675	−0.817	−0.620	−0.184	−0.094	0.261	2849
	Upper-quartile	0.313	−0.707	−0.514	−0.055	0.262	0.039	0.452	
	Median	0.364	−0.404	−0.298	0.017	0.513	0.060	0.504	
	Lower-quartile	0.426	−0.137	−0.171	0.102	0.731	0.084	0.593	
	Max	1.555	2.305	0.265	1.086	3.262	0.244	0.682	
	Positive (%)	100	16.50	2.14	55.39	93.72	98.98	100	
	Negative (%)	0	83.50	97.86	44.59	6.29	1.02	0	
2020	Min	0.299	−1.140	−0.900	−0.195	0.174	−0.045	0.030	2849
	Upper-quartile	0.368	−0.819	−0.395	−0.072	0.306	0.043	0.146	
	Median	0.395	−0.425	−0.241	0.003	0.412	0.062	0.205	
	Lower-quartile	0.440	−0.226	−0.141	0.103	0.630	0.089	0.270	
	Max	0.699	0.260	−0.011	0.692	0.964	0.185	0.358	
	Positive (%)	100	4.84	0	51.07	100	88.84	100	
	Negative (%)	0	95.15	100	48.91	0	11.17	0	

## Data Availability

The data presented in this study are available on request from the corresponding author.

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
