# Peer review of "Spatio-Temporal Variation and Influencing Factors of the Coupling Coordination Degree of Production-Living-Ecological Space in China"

_ijerph, 2022, doi:10.3390/ijerph191610370_

Round 1

Reviewer 1 Report

This manuscript used a classification and evaluation system for PLES, and analyzed the spatiotemporal characteristics, coupling coordination degree, and influencing factors of PLES in China in 2000, 2010 and 2020. This work is a concern for policy makers to build a high-quality national spatial layout and support system for development. Unfortunately, the structure of the manuscript is poor, and there are problems with the manuscript which would need a substantial revision. 

Some suggestions are as follows: 

1. The English needs to be improved. There are many grammar and spelling errors which make the manuscript hard to read. e. g. you’d better make clear how to use past or simple tense. 

2. A key limitation of this study is that what is the novel place compared with previous studies (e. g. references 27, 36)? Why you analyze gravity centers migration trajectory? Did the previous studies have no analysis on influencing factors? 

3. Introduction: The logicality and professionalism of this section need to be strengthened. Reviews on the spatial pattern of production-living-ecological space (PLES) and its influence factor should be expanded. For instance, are there no researches in other countries? The coupled human-earth system is the core of geography study. 

4. Results: The migration trajectories of gravity center are no more than 0.2°, so what is the point of the analysis on gravity center migration trajectory? Some contents would belong to the method section e. g. line 359-363. 

5. Discussion: Most of this section is a summary rather than discussions on findings and what they mean. This section is rather weak. Strengthen explanations in the aspect of the mechanisms, e. g. “gravity centers migration trajectory”, “seriously unbalanced categories”, “influencing factors”.

Reviewer 2 Report

1.This paper explored the coupling coordination degree of production-living-ecological space in China, the structure of the paper is complete, but the paper lacks innovation.

2.In the part of abstract, this paper mentions that there have been few studies on the coupling coordination and influencing factors of PLES at county scale in China. In fact, it has a lot of similar themes according to the Web of Science.The author's division of production-living-ecological space and the methods used are similar to those in the existing papers, especially the coupling coordination method.

3.Compared with OLS model regression results, GWR model could effectively 510 improve model fitting degree. In fact, GTWR model has more advantages than GWR model.

4.Authors should improve the part of literature review.

Reviewer 3 Report

1) The research idea of this study is ordinary and not innovative. And the research methods are not novel and reliable enough to reach the level that can be published.

2) The manuscript is a little difficult to read in its current state and should be formatted for English/grammatical errors.

3) In the section 2.2.1, what’s the meaning of the terms PS, LS, and ES. From formula (1), (2), and (3), I can’t understand the calculation process of P, L, and E, especially the measurement of LA.

4) In the section 2.2.3, authors should explained more clearly about the difference between seriously unbalanced, moderately unbalanced, basically balanced, moderately balanced, and highly balanced in the real world, rather than tell the readers the value.

Reviewer 4 Report

The article "Spatio-temporal variation and influencing factors of the coupling coordination degree of production-living-ecological space in China" is concerned with research and multifunctional, sustainable space development. In particular, it deals with analyses of the spatiotemporal differentiation and coupling coordination characteristics of PLES index at county scale in China using the PLES spatial classification system based on multi-function division of land use. The article is in line with current research trends and to some extent expands knowledge in this area.

I have no comments on the construction and layout of the article. It contains all the elements that a good scientific article should contain.

However, I do have comments on the content presented:

11. The local nature of the research and analysis carried out is very evident. This can be seen especially in the Introduction chapter and what follows in References.  The list of references contains 47 items, of which ONE is not published by Chinese authors and is not directly about China. What I miss here is the reference to international studies. It is possible that PLES is a typically Chinese solution, while please refer to the many articles from around the world that deal with analyses related to sustainable space development. The authors will easily find such elements corresponding to their research in the international literature. Please expand the Introduction chapter by these elements.

22. As I understand from the article, the analyses on PLES are not a novel element of the article. The authors indicate that they conducted county-level studies in China, because such studies have not yet been done (lines 101-112). Where, then, are these counties in the results presented? E.g. lines 234-238 and similarly in the following subsections - one can see the data for China (are they the result of the Authors' calculations, or do they come from other publications?). In a number of the figures presented, one does not see that the distribution of results fits into any administrative boundaries. Some boundaries can be seen in the figures, but they are not explained in the legends to the figures. As it stands, the reader is left with the impression that the authors have done analyses on the same data as other researchers, only sourcing it from a lower administrative level, while the results still apply to the national scale. Please present this in a clear and readable way in the article.

33. The figures presented with the results for each year (this is especially evident in Figures 1, 2 and 3) differ almost nothing. If it were not for the figures entered by the authors in the text, it would be difficult to talk about any changes here at all. Please refer to the scale of these changes in the conclusions (lines 491-500).

44. If, as the Authors write, research in this area was carried out in China please refer to the results in the Disscussion section. As it stands, I lack justification that the use of county-level data, as indicated by the Authors, did anything really change. Are the results obtained consistent with other studies conducted on a broader national or regional scale? In the Disscussion chapter, I also miss references to the results. obtained in this type of analysis in other countries (see comment 1).

      The article is interesting and well researched. However, it lacks a broader global context for me. Please also refer to the above comments. The article needs some supplementation and modification.

Round 2

Reviewer 2 Report

Moderate English changes required

Reviewer 3 Report

The authors have addressed most of my concerns, and I recommend it to be accepted for publication.